# *Ganoderma lucidum* Spore Powder Alleviates Metabolic-Associated Fatty Liver Disease by Improving Lipid Accumulation and Oxidative Stress via Autophagy

**DOI:** 10.3390/antiox13121501

**Published:** 2024-12-09

**Authors:** Yuxuan Zhang, Jiali Zhou, Lan Yang, Hang Xiao, Dongbo Liu, Xincong Kang

**Affiliations:** 1Horticulture College, Hunan Agricultural University, Changsha 410128, China; 2State Key Laboratory of Subhealth Intervention Technology, Hunan Agricultural University, Changsha 410128, China; 3Hunan Provincial Engineering Research Center of Medical Nutrition Intervention Technology for Metabolic Diseases, Hunan Agricultural University, Changsha 410128, China; 4School of Pharmacy, North Sichuan Medical College, Nanchong 637000, China; 5Department of Food Science, University of Massachusetts, Amherst, MA 01003, USA; 6National Research Center of Engineering Technology for Utilization of Botanical Functional Ingredients, Hunan Agricultural University, Changsha 410128, China

**Keywords:** lipid deposition, antioxidant, *Ganoderma lucidum* spores, water extract, autophagy

## Abstract

Lipid accumulation and oxidative stress, which could be improved by autophagy, are the “hits” of metabolic-associated fatty liver disease (MAFLD). *Ganoderma lucidum* spore powder (GLSP) has the effect of improving liver function. However, there are few reports about its effects on and mechanisms impacting MAFLD alleviation. This study investigated the effect of GLSP on hepatic lipid accumulation and oxidative stress and explored the role that autophagy played in this effect. The results showed that GLSP effectively reduced lipid accumulation and activated autophagy in the livers of mice with high-fat-diet-induced disease and palmitic acid-induced hepatocytes. GLSP reduced the lipid accumulation by reducing lipogenesis and promoting lipid oxidation in HepG2 cells. It decreased the production of ROS, increased the activity of SOD and CAT, and improved the mitochondrial membrane potential via the Keap1/Nrf2 pathway. The alleviating effects of GLSP on the lipid accumulation and oxidative stress was reversed by 3-methyladenine (3-MA), an autophagy inhibitor. GLSP activated autophagy via the AMPK pathway in HepG2 cells. In conclusion, GLSP could attenuate MAFLD by the improvement of lipid accumulation and oxidative stress via autophagy. This paper is the first to report the improvement of MAFLD through autophagy promotion. It will shed novel light on the discovery of therapeutic strategies targeting autophagy for MAFLD.

## 1. Introduction

Metabolic-associated fatty liver disease (MAFLD), renamed from non-alcoholic fatty liver disease (NAFLD) in 2020, is the most prevalent chronic liver disease, with a global prevalence of 39.22% [1]. It develops from hepatic steatosis (fatty liver) and one of the following metabolic conditions: metabolic dysregulation, overweight/obesity, or type 2 diabetes [2]. Up to now, there is no effective approved drug for MAFLD treatment.

The pathogenesis of MAFLD is complicated, and the most widely accepted theory is “multiple parallel hits” proposed for NAFLD [3]. According to this theory, lipid accumulation is “the first hit”, which increases the vulnerability of liver to many other risk factors including oxidative stress. Excessive hepatic lipid accumulation causes the production of lipotoxins that induce mitochondrial dysfunction and the over-production of reactive oxygen species (ROS) [4]. In excess, ROS induce lipid peroxidation that results in the synthesis of reactive lipids [4]. The reactive lipid derivatives will further amplify liver damage by promoting the release of ROS outside the hepatocytes [5]. In a word, MAFLD typically arises from aberrant lipid metabolism and oxidative stress.

Autophagy, a non-invasive response to internal and external stimuli, improves lipid metabolism disorders and maintains lipid metabolism homeostasis by degrading intracellular lipid droplets [6]. At the same time, it is essential to remove dysfunctional mitochondria, thus limiting the production of ROS under the conditions of nutrient overload [7]. Autophagic and lysosomal functions are impaired in livers in patients with NAFLD [8]. The suppression of autophagy, resulting in the accumulation of inactive mitochondria and elevation of oxidative stress, consequently impairs the physiological turnover of hepatic cytoplasm [9]. Thus, interventions that target autophagy might be a novel strategy for MAFLD treatment.

The *Ganoderma lucidum* spore (GLS) is the mature germ cell of *G. lucidum*, which is a medicinal edible fungus and has been used in China to promote health and longevity for centuries. *G. lucidum* exhibits a broad range of hepatoprotective impacts in various liver disorders, such as non-alcoholic and alcohol-induced fatty liver disease, hepatitis B, hepatic fibrosis, and hepatic cancer. The GLS has the effect of improving liver function [10]. A GLS administered by gavage could inhibit CCl_4_-induced liver fibrosis, alleviate dibutyl phthalate and benzo(a)pyrene-induced liver injury, and attenuate sub-acute alcoholic hepatic injury [11,12]. In addition, the GLS could protect mice against high-fat-intake-induced obesity by regulating lipid metabolism [13]. The GLS has been followed with interest and commercialized as a dietary supplement due to the development of technology for breaking the spore wall [10]. However, knowledge of the pharmacology and mechanism of the GLS in MAFLD is limited.

The aim of this study was to explore the effects of *G. lucidum* spore powder (GLSP) on MAFLD and elucidate its mechanism. We evaluated the potential of GLSP to regulate hepatic lipid metabolic homeostasis and improve autophagy levels in mice and subsequently investigated its effects and autophagy-centered mechanism on lipid accumulation and oxidative stress in HepG2 cells. This study will provide a theoretical basis and mechanism for the exploration of GLSP to ameliorate MAFLD and will shed novel light on the discovery of therapeutic strategies targeting autophagy for MAFLD.

## 2. Materials and Methods

### 2.1. Chemicals and Reagents

Dualbecco’s modified Eagle’s medium (DMEM), trypsin/EDTA, and penicillin-streptomycin were purchased from Gibco (Grand Island, NY, USA). Fetal bovine serum (FBS) was purchased from Biological Industries (Beit-Haemek, Akko, Israel). Bovine serum albumin (BSA) was purchased from Beyotime Biotechnology (Shanghai, China). The primary antibodies against acyl coenzyme A oxidase 1 (ACOX1), the sterol regulatory element binding protein (SREBP1), the peroxisome proliferator-activated receptor alpha (PPARα), acetyl-CoA carboxylase (ACC1), P-ACC1, and carnitine palmitoyltransferase 1 A (CPT1A) were purchased from Proteintech (Wuhan, China). The primary antibodies against fatty acid synthase (FASN), Kelch-like ECH-associated protein 1 (Keap1), and nuclear factor erythroid 2-related factor 2 (Nrf2) were purchased from Aifang biological (Changsha, China). The primary antibody against heme oxygenase 1 (HO-1) was purchased from Beyotime Biotechnology (Shanghai, China). The antibody against the protein light chain 3 (LC3) was purchased from Medical & Biological Laboratories Co., Ltd. (Tokyo, Japan). The primary antibodies against AMPK, P-AMPK, mTOR, P-mTOR, goat anti-mouse IgG, goat anti-rabbit IgG, and anti-rabbit IgG Alexa Fluor 488 conjugate were purchased from Cell Signaling technology (Beverly, MA, USA).

### 2.2. Preparation of GLSP

An aqueous extract of 20.0 g of broken GLSP (XPD010-2013, State Key Laboratory of Subhealth Intervention Technology, Changsha, China) was extracted at a solid–liquid ratio of 1:20 at 60 °C for 2 h. The filtrate solution was concentrated and freeze-dried to obtain a crude aqueous extract which was mixed with medium at a concentration of 64 mg/mL and stored at −20 °C for later use. The total polysaccharide content was measured by the phenol–sulfuric acid colorimetric method, and the total triterpene content was measured by the vanillin–sulfuric acid color method according to the Chinese Pharmacopoeia.

### 2.3. Preparation of Palmitic Acid (PA)

PA-BSA conjugates were prepared by PA saponification with NaOH and mixing with BSA. In short, 0.307 g of PA was dissolved in a 0.1 M NaOH solution at 75 °C for 30 min for full saponification and finally became colorless, clear, and transparent. The same volume of PA solution was quickly added to the 40% BSA solution, and 20 mM of PA was obtained. The storage solution was sterilized by filtration and could be stored at 4 °C for a long time.

### 2.4. Animals

Five-week-old male C57BL/6 mice were purchased from Beijing Viatal River Laboratory Animal Technology Co., Ltd. (License: SCKK (Jing) 2019-0009, SYXK (Jing) 2017-0033). The animal experiment was carried out in accordance with China Animal Welfare Legislation and was approved by the Experimental Animal Ethics Committee of South China Agricultural University (2021b171). The mice were housed individually under temperature- (22 ± 5 °C), humidity- (50 ± 5%), noise- (≤60 dB) and light- (12 h light/dark cycle) controlled conditions with access to water ad libitum and a diet. After acclimating for 1 week, the mice were randomly allocated to receive a control diet (containing 10% fat, D12450, Research Diets Inc., New Brunswick, NJ, USA) or a high-fat diet (60% fat, D12492, Research Diets Inc.). The mice were divided into the following four groups (*n* = 12 in each group). (1) The control group (NCD group): the mice were fed with a normal mouse diet for 16 weeks. (2) The HFD group: the mice were fed with a high-fat rat diet (HFD) for 16 weeks. (3) The SIM group: the mice were fed with a high-fat rat diet containing simvastatin (SIM) at a 15 mg/kg b.w/day concentration for 16 weeks. SIM is an oral hypolipidemic agent, and it is the 2nd most widely prescribed lipid-lowering drug. (4) The GLSP group: the mice were fed with a high-fat diet containing 3% GLSP for 16 weeks. The mice’s body weight and food intake (*n* = 12 in each group) were monitored once a week over a 16-week course of treatment. At the end of the 16 weeks, the mice were euthanized with an intraperitoneal injection of 10% chloral hydrate solution (3 mL/kg). Liver and serum samples from each mouse were collected for measurement. The livers were weighed (*n* = 12 in each group) and the liver index was calculated by the following formula: liver index (mg/g) = weight of liver/weight of body. The total cholesterol (TC) and triglyceride (TG) levels in the serum and liver (*n* = 6 in each group) were measured with kits from Nanjing Jiancheng Bioengineering Institute (Nanjing, China) according to the manufacturer’s protocol.

### 2.5. Histological Analysis and Immunohistochemical Staining of Liver Tissue

Liver tissue samples from each mouse (*n* = 3 in each group) were fixed in 10% phosphate-buffered formalin and then embedded in paraffin blocks. The paraffin-embedded liver tissues were stained with hematoxylin and eosin (H&E) for the histological examination of the mice livers. Frozen liver pieces were stained with Oil Red O to assess liver lipid accumulation. Immunohistochemistry was applied to detect the protein expression of the autophagy biomarker LC3 in the liver tissues. For antigen retrieval using citric acid buffer, the slides were heated at 120 °C for 20 min and then cooled for 20 min at room temperature. The slides were incubated overnight with the primary antibody LC3 and were then incubated with peroxidase-conjugated streptavidin for 15 min. Diaminobenzidine–tetrahydrochloride was used as the substrate to detect antigen–antibody binding. All sections were scanned using 3D Histech (Budapest, Hungary).

### 2.6. Cell Culture and Treatments

Human hepatic HepG2 cells (Cell Bank of the Shanghai Institute of Biochemistry and Cell Biology, Chinese Academy of Sciences, Shanghai, China) were cultured in DMEM/high glucose (4.5 g/L) containing 10% FBS and a 1% penicillin–streptomycin solution. The cells were grown in a humidified incubator at 37 °C with 5% CO_2_. The HepG2 cells were treated with PA to mimic lipotoxicity conditions in vitro. To investigate the effect of GLSP on lipid accumulation and oxidative stress, the attached cells were treated with GLSP (50, 200, 800 μg/mL) for 12 h and then exposed to 0.6 mM PA for 12 h. To determine whether GLSP improves MAFLD by autophagy, the HepG2 cells were treated with 3-methyladenine (3-MA) (3 mM) and GLSP (800 μg/mL) for 12 h and then exposed to 0.6 mM PA for 12 h. Oil Red O staining, triglyceride (TG) levels, and total cholesterol (TC) levels were detected to measure lipid accumulation in the HepG2 cells. The ROS, antioxidant enzymes, and mitochondrial membrane potential were measured to analyze the effect of the GLSP on oxidative stress. Immunofluorescence and Western blotting were employed to elucidate the molecular mechanism of the GLSP on alleviating MAFLD.

### 2.7. Cell Viability

Cell viability was assessed by CCK-8 assay. Briefly, HepG2 cells (1 × 10^4^/well) were cultured in a 96-well plate and incubated with different concentrations of GLSP (0, 200, 400, 800, 1600, 3200 μg/mL) for 12 h. After the wells were exposed to 0.6 mM PA for 12 h, 10 μL of CCK-8 (Beyotime Biotechnology, Shanghai, China) reagent was added to each well. The OD value was measured using a microplate reader at a 450 nm wavelength after 1 h.

### 2.8. Oil Red O Staining

Cells were inoculated in a 24-well plate with an appropriate density and then incubated with GLSP, 3-MA, and PA for a certain period of time. The cells were fixed with 4% paraformaldehyde for 10 min, followed by staining with Oil Red O solution (Solarbio, Beijing, China) for 30 min in the dark at 37 °C. The stained cells were washed 3 times with phosphate-buffered saline (PBS). After washing again, the cells were observed with an inverted microscope Zeiss LSM710 (Zeiss, Dusseldorf, Germany) and photographed.

### 2.9. Triglyceride and Total Cholesterol

Cells were inoculated in a 6-well plate with an appropriate density and then incubated with GLSP, 3-MA, and PA for a certain period of time. The cells were collected and treated according to the protocols specified in the triglyceride assay kit (Nanjing Jiancheng Bioengineering Institute, Nanjing, China) and total cholesterol assay kit (Nanjing Jiancheng Bioengineering Institute, Nanjing, China). The protein concentration was calculated using the Bicinchoninic Acid (BCA) Protein Assay Kit (Beyotime Biotechnology, Shanghai, China), and, finally, the TG and TC contents were calculated.

### 2.10. Reactive Oxygen Species (ROS)

To explore the effect of GLSP on oxidative stress, the level of ROS in the HepG2 cells was measured by DCFH-DA according to the manufacturer’s instructions in the ROS Assay Kit (Beyotime Biotechnology, Shanghai, China). The cells were inoculated in a 24-well plate with an appropriate density (1 × 10^4^/well) and then incubated under different conditions: ① with GLSP (50, 200, 800 μg/mL) for 12 h, ② with 800 μg/mL GLSP, and 3 with mM 3-MA for 12 h followed by exposure to 0.6 mM PA for 12 h. The cells were washed with serum-free medium and cultured in 10 μM DCFH-DA for 30 min at 37 °C and then observed using a fluorescence-inverted microscope and photographed.

### 2.11. Antioxidant Enzyme

Cells were inoculated in a 6-well plate with an appropriate density and then incubated with GLSP, 3-MA, and PA for a certain period of time. The activities of superoxide dismutase (SOD) and catalase (CAT) in the cells were measured according to the manufacturer’s instructions in the Total Superoxide Dismutase Assay Kit with WST-8 (Beyotime Biotechnology, Shanghai, China) and the CheKine™ Micro Catalase Activity Assay Kit (Abbkine Biotechnology, Wuhan, China).

### 2.12. Measurement of Mitochondrial Membrane Potential

The level of the mitochondrial membrane potential was measured according to the instructions in the Mitochondrial membrane potential assay kit (Beyotime Biotechnology, Shanghai, China). Cells were inoculated in a 6-well plate with an appropriate density (1 × 10^4^/well) and then incubated under different conditions: ① with GLSP (50, 200, 800 μg/mL) for 12 h, ② with 800 μg/mL GLSP, and 3 mM 3-MA for 12 h followed by exposure to 0.6 mM PA for 12 h. After washing with PBS, the HepG2 cells were incubated with JC-1 at 37 °C for 30 min in the dark. After incubation, the cells were washed with JC-1 staining buffer three times, observed using a fluorescence-inverted microscope, and photographed.

### 2.13. Immunofluorescence

HepG2 cells were plated on 10 mm glass coverslips in a 24-well plate and then incubated with GLSP, 3-MA, and PA for a certain period of time. The cells were fixed with 4% paraformaldehyde for 10 min and then washed with PBS three times. After being treated with blocking solution (PBS with 10% normal goat serum) for 30 min, the cells were incubated at 37 °C for 1 h with a primary antibody, the LC3 polyclonal antibody. Then, the cells were subsequently incubated at 37 °C for 1 h with a secondary antibody, the anti-rabbit IgG Alexa Fluor 488 conjugate. Imaging was performed using a fluorescence-inverted microscope, and the samples were photographed.

### 2.14. Western Blot

Cells were inoculated in a 6-well plate with an appropriate density and then incubated with GLSP, 3-MA, and PA for a certain period of time. After incubation, the cells were lysed in an appropriate amount of cell lysis buffer, and the lysates were heated at 95 °C for 10 min. Next, loading buffer was added and mixed well. This mixture was further heated at 95 °C for another 10 min to obtain the cell protein sample. An equal number of proteins (20–40 μg/condition) were resolved by SDS-PAGE before being transferred to a polyvinylidene difluoride membrane. The membrane was blocked with QuickBlock™ Blocking Buffer for Western Blot (Beyotime Biotechnology, Shanghai, China) at room temperature for 10–15 min. Different bands were cut out according to the molecular weight of the target protein, and the corresponding primary antibody solution (1:1000) was used for incubation overnight at 4 °C. The membrane was then incubated with an appropriate secondary antibody tagged with horseradish peroxidase. The Chemiluminescence and fluorescence Image System (Thermo Fisher Scientific, Singapore) was used to observe the expression of the target protein.

### 2.15. Quantification and Statistical Analysis

The experimental data were analyzed by GraphPad Prism 9 software, and the pictures were sorted by Adobe Illustrator 2023 (v27.0, Adobe Inc., Atlanta, GA, USA) software. *p*-values of less than 0.01 or 0.05 were considered statistically significant.

## 3. Results

### 3.1. GLSP Alleviated High-Fat-Induced MAFLD in Mice

A significant increase in the body weight of the HFD-fed mice was observed from 1 to 12 weeks, as compared to those fed with NCD, although the food intake was not significantly different (Figure 1A–C). A characteristic manifestation of MAFLD is abnormal hepatic lipid accumulation, which is mainly attributed to the imbalance of lipid metabolism in the hepatocytes. The livers of the mice in the NCD group were dark red, while the livers of the HFD-fed mice lost their blood color and looked light yellow and pale in color, which is a typical and intuitive manifestation of hepatic steatosis (Figure 1D). After GLSP supplementation, although the food intake was increased, the HFD-induced body weight gain was notably inhibited from the 11th week, earlier than the positive control SIM group. The liver weight was markedly reduced in the GLSP group, and the dark brown–red color of the liver was recovered which looked like in the SIM group (Figure 1D–F).

In the NCD group, the hepatic lobule structure of the mice was intact and the hepatocytes were uniform in size, while in the HFD group, the hepatic lobule structure was disrupted, and the hepatocytes exhibited irregular morphology with numerous vacuoles, indicating severe hepatic steatosis (Figure 1G). Consistent with the H&E staining results, the Oil Red O staining showed the presence of abundant lipid droplets in the liver of the HFD group, indicating the successful establishment of the mouse model of MAFLD (Figure 1G). Compared to the HFD group, the SIM and GLSP groups showed significant improvement in their hepatic lobule structure, with a notable reduction in the number of lipid droplets and vacuoles. Additionally, the levels of TG and TC in the liver and serum of the HFD-group mice were significantly higher than those in the NCD-group mice. Compared to the HFD-group mice, the SIM- and GLSP-group mice showed significant improvements in their TG and TC levels (Figure 1H). These results suggest that the GLSP could ameliorate the lipid accumulation and hepatic steatosis induced by a high-fat diet.

Polysaccharides and triterpenoids have been recognized as primary bioactive ingredients in *G. lucidum* spore powder. Therefore, we detected the total polysaccharides and triterpenoids contents of the GLSP. The results showed that it contained 48% of total polysaccharides and 3.8% of total triterpenoids. Polysaccharides are primarily composed of four monosaccharides: glucose (86.62%), galactose (7.11%), mannose (4.48%), and fucose (1.79%), which we have explored in another study [14].

### 3.2. GLSP Alleviated PA-Induced Hepatic Lipid Accumulation by Reducing Lipogenesis and Promoting Lipid Oxidation

To investigate the mechanism of GLSP alleviating MAFLD, we assessed the effects of GLSP in a PA-induced MAFLD cell model in HepG2 cells. GLSP with a concentration of 0 to 3200 µg/mL had no significant effect on cell viability (Figure 2A). The lipid droplets in the HepG2 cells significantly increased after PA induction, while they decreased in a concentration-dependent manner after GLSP intervention (Figure 2B). The TG and TC levels in the PA group were higher than those in the control group, whereas the GLSP treatment markedly reduced the TG and TC levels (Figure 2C,D). These results suggest that the GLSP supplementation suppressed lipid accumulation in the hepatic cells.

The protein levels of the lipid synthesis proteins ACC, SREBP1, and FASN and the fatty-acid oxidation-related proteins PPARα, CPT1A, and ACOX1 were detected to verify the pathway used by the GLSP to reduce lipid accumulation in the HepG2 cells. After PA induction, the protein levels of SREBP1 and FASN were significantly increased, while the expression of phosphorylated ACC, PPARα, CPT1A, and ACOX1 were significantly reduced. Treatment with GLSP could reverse the effect of PA, reducing the expression of lipogenic protein and increasing the levels of lipid oxidation-related proteins (Figure 2E). These results indicate that GLSP can alleviate PA-induced liver lipid accumulation by reducing lipogenesis and promoting lipid oxidation.

### 3.3. GLSP Improved PA-Induced Oxidative Stress Through Keap1-Nrf2 Pathway Induced by PA

Compared to the control group, the level of ROS in the PA group increased significantly. When the cells were treated with 50~800 µg/mL of GLSP, intracellular ROS formation was suppressed in a dose-dependent manner compared to that of the PA group (Figure 3A). The PA significantly reduced the activities of SOD and CAT, while GLSP increased such activities (Figure 3B,C).

When oxidative stress occurs, the excess production of ROS leads to mitochondrial dysfunction, characterized by the loss of the mitochondrial membrane potential. Therefore, we evaluated the mitochondrial membrane potential by JC-1 assay. When the mitochondrial membrane potential is high, JC-1 aggregates in the matrix of mitochondria, forming polymers (J-aggregates) that produce red fluorescence. When the mitochondrial membrane potential is low, JC-1 cannot aggregate in the matrix of mitochondria and exists as monomers, producing green fluorescence. Hence, the change in fluorescence color can be conveniently used to detect the variation in the mitochondrial membrane potential. In this study, PA decreased the red fluorescence level and increased the green fluorescence level in HepG2 cells, while GLSP reversed this change (Figure 3D). The results showed that the GLSP could improve the mitochondrial membrane potential damage induced by the PA. This suggests that GLSP improves oxidative stress by restoring the mitochondrial membrane potential. The Keap1-Nrf2 pathway is one of the most important defense mechanisms against oxidative stress. The expressions of Keap1, Nrf2, and HO-1 in the HepG2 cells were detected by Western blotting. After PA induction, the protein levels of Keap1 were significantly increased, while the protein levels of Nrf2 and HO-1 were decreased. Treatment with GLSP could down-regulate Keap1 and up-regulate the expression of Nrf2 and HO-1 (Figure 3E). These results suggest that GLSP can improve oxidative stress through the Keap1-Nrf2 pathway.

### 3.4. GLSP Induced Autophagy via AMPK Signal Pathway

To verify the role of autophagy in MAFLD alleviation, we detected the expression of LC3 in the mouse liver through immunohistochemistry. In the NCD group, most areas of the mouse liver tissue were stained brown, indicating the specific binding of the LC3 antibody, while the brown-stained area was significantly reduced in the HFD group (Figure 4A). With the intervention with GLSP, the brown-stained area was notably increased compared to the HFD group. This suggests that a high-fat diet inhibited cellular autophagy in the mouse liver, while GLSP could improve the suppressed autophagy caused by a high-fat diet (Figure 4A).

In the HepG2 cells, after treatment with PA for 12 h, the number of autophagosomes was reduced significantly compared to that in the control group (Figure 4B) and the ratio of LC3-II/LC3-I was remarkably lower (Figure 4C). When the cells were treated with 200 and 800 µg/mL of GLSP, the number of autophagosomes was increased compared to that in the PA-treated group (Figure 4B). As an upstream protein of autophagy, the p-AMPK to AMPK ratio was significantly increased, and the p-mTOR to mTOR ratio was significantly decreased after being treated with 200 and 800 µg/mL of GLSP (Figure 4C). The results indicate that GLSP could alleviate the autophagy of HepG2 cells inhibited by PA through the AMPK pathway.

### 3.5. GLSP Improved Lipid Accumulation in HepG2 Cells Dependent on Autophagy

To explore whether GLSP improves MAFLD through autophagy, the autophagy inhibitor 3-MA was added to co-culture with GLSP. In the 3-MA-treated group, the number of autophagy fluorescence spots was reduced, the ratios of LC3 II to LC3 I and p-AMPK to AMPK were decreased, and the ratio of p-mTOR to mTOR was increased compared to the 800 µg/mL GLSP-treated group (Figure 5A,B), indicating that the 3-MA blocked autophagy successfully. The decreased lipid droplets and TG and TC levels in the GLSP intervention were abolished by 3-MA (Figure 5C–E). The expressions of the lipid synthesis protein FASN, SREBP1, and the activated form of ACC were increased, while that of the fatty-acid oxidation-related proteins ACOX1, CPT1A, and PPARα were decreased after the 3-MA addition compared to the GLSP group (Figure 5F). In a word, the relieving effect of GLSP on lipid accumulation in HepG2 cells is dependent on autophagy.

### 3.6. Autophagy Contributed to Effect of GLSP Protecting HepG2 Cells from Oxidative Stress Under PA Stress

Compared to that in the GLSP group, the level of ROS in the 3-MA group increased significantly (Figure 6A). In addition, the effect of GLSP on the activity of SOD and CAT was reversed by 3-MA (Figure 6B,C). Meanwhile, compared to the GLSP group, the 3-MA treatment decreased the red fluorescence level and increased the green fluorescence level in HepG2 cells (Figure 6D). This means that the protective effect of GLSP against mitochondrial membrane potential injury is abrogated by 3-MA. The protein expression level of Keap1, Nrf2, and HO-1 proved that 3-MA blocked the GLSP-improved effect on oxidative stress (Figure 6E). In conclusion, 3-MA reverses the effect of GLSP on improving and alleviating PA-induced oxidative stress.

## 4. Discussion

At present, the high prevalence of MAFLD is attracting increasing attention. However, there is still a lack of effective treatment strategies. In the present study, we found that GLSP could effectively alleviate MAFLD and activate autophagy in mice. It suppressed PA-induced lipid accumulation and oxidative stress in HepG2 cells. It reduced lipid accumulation by decreasing de novo lipogenesis and increasing lipid β-oxidation and improved oxidative stress by activating the Keap1-Nrf2 pathway. All these protective effects against MAFLD via autophagy.

The protective effects of GLSP against MAFLD may be due to the polysaccharides and triterpenoids it contains. Polysaccharides and triterpenoids are two main active biological components in the aqueous extract of GLSP [15,16]. *G. lucidum* polysaccharides are reported to ameliorate hepatic steatosis and oxidative stress in vivo and in vitro [17,18]. In our previous study, *G. lucidum* polysaccharides could ameliorate MAFLD by inhibiting lipid accumulation and alleviating oxidative stress in HepG2 cells [14]. *G. lucidum* triterpenoids, the main active ingredients for anti-liver fibrosis [19], is also reported to regulate lipid metabolism and alleviate oxidative stress [20]. In the future, we will purify the triterpene monomer in GLSP and explore its biological activities and mechanisms.

GLSP reduces lipid production and promotes fatty-acid oxidation via the AMPK pathway, thereby reducing intracellular fat deposition. The most direct cause of MAFLD is the imbalance of lipid metabolism, which leads to excessive lipid accumulation in the liver [21]. The rate of lipid accumulation is governed by AMPK, a key protein for the regulation of hepatic lipid metabolism [22,23]. The inhibition of AMPK increases the expression of SREBP-1c, which in turn induces lipogenic proteins, including ACC and FASN, and produces saturated fatty acids and monounsaturated fatty acids. Activated AMPK phosphorylates ACC to inhibit the synthesis of malonyl-CoA, which inhibits mitochondrial CPT-1, consequently facilitating fatty acid uptake into mitochondria for fatty-acid β-oxidation. In addition, AMPK enhances the activity of PPARα, a key transcription factor of lipid metabolism mainly restricted to the liver, thereby increasing the expression of lipid oxidation enzymes (e.g., ACOX1), reducing lipid deposition, and consequently inhibiting inflammation [24].

How does GLSP activate AMPK? According to previous reports, AMPK can be activated via indirect and direct ways [25]. In the indirect way, AMPK could be indirectly activated by increasing the AMP/ADP:ATP ratio or calcium accumulation (e.g., quercetin, ginsenoside Rb1, α-lipoic acid, and cryptotanshinone) [25]. High levels of AMP and ADP bind to the CBS3 (cystathionine β-synthase 3) of the AMPK γ-subunit, which not only prevents the phosphatases from accessing the T172 of the AMPK α-subunit to increase phosphorylation but also stimulates LKB1 (liver kinase B1)-mediated phosphorylation, consequently activating AMPK [26]. In addition, the binding of AMP, but not of ADP, to CBS1 increases intrinsic AMPK activity by inducing its allosteric activation [26]. Intracellular calcium activates AMPK through CaMKK2 (calcium-/calmodulin-dependent kinase kinase 2)-mediated phosphorylation [26]. In the direct way, AMPK could be activated by directly binding to the subunit of AMPK (e.g., 5-Aminoimidazole-4-carboxamide riboside (AICAR), thienopyridone (A-769662), etc.), inducing conformation changes in the AMPK complex [25]. It is reported that ganodermanondiol extracted from *G. lucidum* could enhance the activation/phosphorylation of AMPK and its upstream kinase activators LKB1 and Ca^2+^/calmodulin-dependent protein kinase-II (CaMKII) in HepG2 cells [27]. The abundant hydroxyl groups in polysaccharide moieties could bind to the CBM (carbohydrate binding module) motif in the β subunit of AMPK [28]. Therefore, the function of GLSP in activating AMPK may be attributed to the ganodermanondiol and polysaccharides in the water extract of GLSP via indirect and direct action.

GLSP can improve oxidative stress and mitochondrial dysfunction by activating the Keap1-Nrf2 pathway. In the NAFLD multiple parallel strike theory, oxidative stress is considered to be one of the main causes of liver injury and disease progression [29]. In normal conditions, free fatty acids are oxidized to carbon dioxide and water, and lipid content is thereby reduced in the liver [5]. When the lipids overload, they will lead to chronic mitochondrial dysfunction characterized by excessive electron leakage, which consequently results in ROS overproduction and oxidative stress [30]. ROS overproduction induces hepatocyte cell death and cause lipid peroxidation that results in the synthesis of reactive lipids. The Keap1-Nrf2 pathway is the principal protective response to oxidative and electrophilic stresses. When cells are attacked by ROS or electrophiles, Nrf2 dissociates from Keap1 and translocates rapidly into the nucleus; it first binds to small MAF proteins to form heterodimers and then binds to antioxidant response elements and finally activates the expression of the antioxidant enzymes HO-1, SOD, and CAT [29].

GLSP is directly proved to target autophagy to improve PA-induced lipid accumulation and oxidative stress. Recently, the importance of autophagy in regulating hepatic lipid metabolism has been acknowledged, and it has also been observed that impaired autophagy plays an important role in the development of NAFLD [31]. Promoting autophagy could alleviate high-fat-diet-induced NAFLD [32]. The dysregulation of hepatic lysosomal/autophagy has been linked to pathogenic steatosis under the condition of NAFLD [33]. Thyroid hormone, hydrogen sulfide, and DA-1241 are reported to ameliorate NAFLD and induce autophagy [34,35,36]. Thyroid hormone improves non-alcoholic steatohepatitis, an advanced form of MAFLD, by simultaneously restoring autophagy, increasing mitochondrial biogenesis, and attenuating oxidative stress in mice. H2S reverses the increased body weight, liver weight, and liver TC and TG content in HFD-fed mice and restores the impaired hepatic autophagic flux. DA-1241 reduces triglyceride content in the liver and blocks autophagic flow in HepG2 cells. However, all these investigations only point out that autophagy occurs simultaneously with remission. They have not directly proved whether the remission effect is via autophagy and have not illustrated the role of autophagy in lipid accumulation and oxidative stress. In this study, we proved that GLSP could restore autophagy, consequently improving lipid accumulation and oxidative stress, leading to MAFLD alleviation.

## 5. Conclusions

In conclusion, GLSP can alleviate MAFLD in vivo and in vitro and target autophagy to improve lipid accumulation and oxidative stress. This is the first report on the improvement of MAFLD through the promotion of autophagy. This study provides a theoretical basis and mechanism of exploration for the amelioration of MAFLD through GLSP and suggests that autophagy could be an effective target for MAFLD therapy.

## Figures and Tables

**Figure 1 antioxidants-13-01501-f001:**
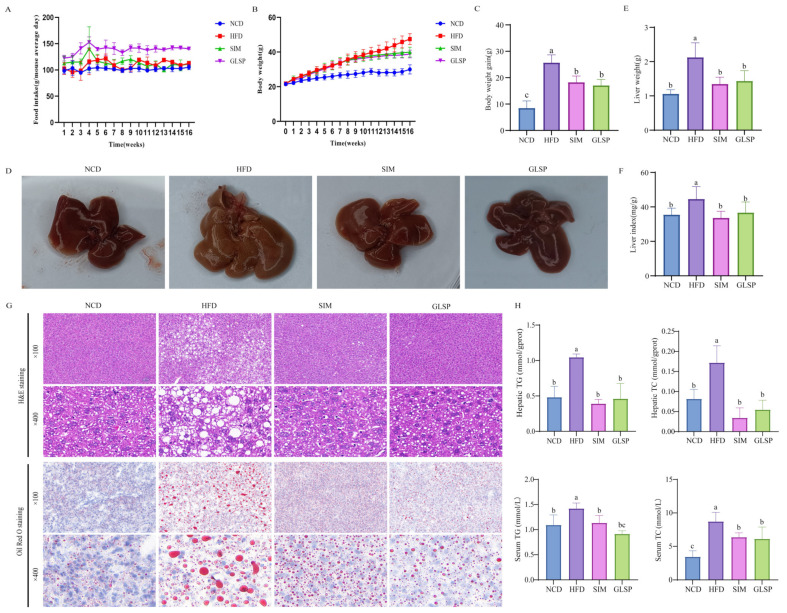
GLSP alleviated high-fat-induced MAFLD in mice. (**A**) Food intake (*n* = 12 in each group), (**B**) body weight (*n* = 12 in each group), and (**C**) body weight gain (*n* = 12 in each group); (**D**) representative images of livers; (**E**) liver weight (*n* = 12 in each group); (**F**) liver index (weight of liver/weight of body; *n* = 12 in each group); (**G**) representative images of H&E and Oil Red O staining of liver; (**H**) TG and TC levels in liver and serum (*n* = 6 in each group). NCD: control group; HFD: high-fat diet group; SIM: simvastatin (15 mg/kg b.w/day) group; GLSP: GLSP (3%) group. Data are expressed as mean ± SEM, and different letters indicate significant differences (*p* < 0.05). Statistical differences were assessed by Tukey’s test of one-way ANOVA.

**Figure 2 antioxidants-13-01501-f002:**
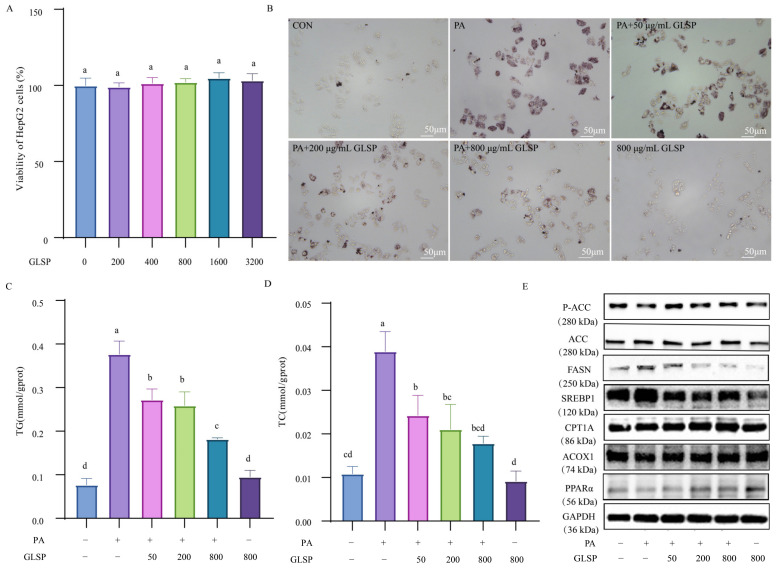
GLSP reduced palmitic acid (PA)-induced lipid accumulation in HepG2 cells. (**A**) Effects of 0, 200, 400, 800, 1600, and 3200 μg/mL of GLSP on cell viability after 12 h treatment. (**B**) Intracellular lipid accumulation was analyzed by Oil red O staining; scale bars, 50 μm. (**C**,**D**) Intracellular TG and TC levels in HepG2 cells. (**E**) Western blot analysis of ACC, FASN, SREBP1, ACOX1, CPT1A, PPARα, and GAPDH protein levels. HepG2 cells treated with or without PA (0.6 mM) and with or without GLSP (50, 200, 800 μg/mL). Data are expressed as mean ± SEM, and different letters indicate significant differences (*p* < 0.05). Statistical differences were assessed by Tukey’s test of one-way ANOVA.

**Figure 3 antioxidants-13-01501-f003:**
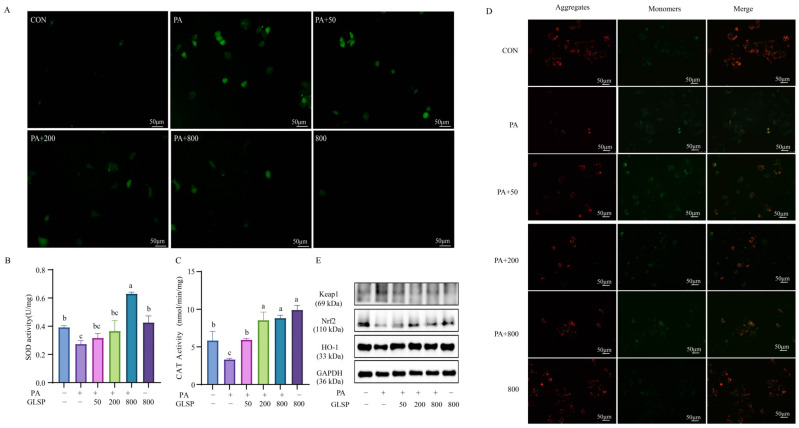
Effect of GLSP on PA-induced oxidative stress in HepG2 cells. (**A**) ROS production detected by DCFH-DA detector; scale bars, 50 μm. (**B**) SOD activity was measured with Total Superoxide Dismutase Assay Kit with WST-8. (**C**) CAT activity was measured with Micro CAT Assay Kit. (**D**) Effect of GLSP on mitochondrial damage. GLSP was added into HepG2 cells for 12 h. Representative images of JC-1-derived red and green fluorescence; scale bars, 50 μm. (**E**) Western blot analysis of Keap1, Nrf2, and HO-1 protein levels with GAPDH as control. HepG2 cells treated with or without PA (0.6 mM) and with or without GLSP (50, 200, 800 μg/mL). Data are expressed as mean ± S.E., and different letters indicate significant differences (*p* < 0.05). Statistical differences were assessed by Tukey’s test of one-way ANOVA.

**Figure 4 antioxidants-13-01501-f004:**
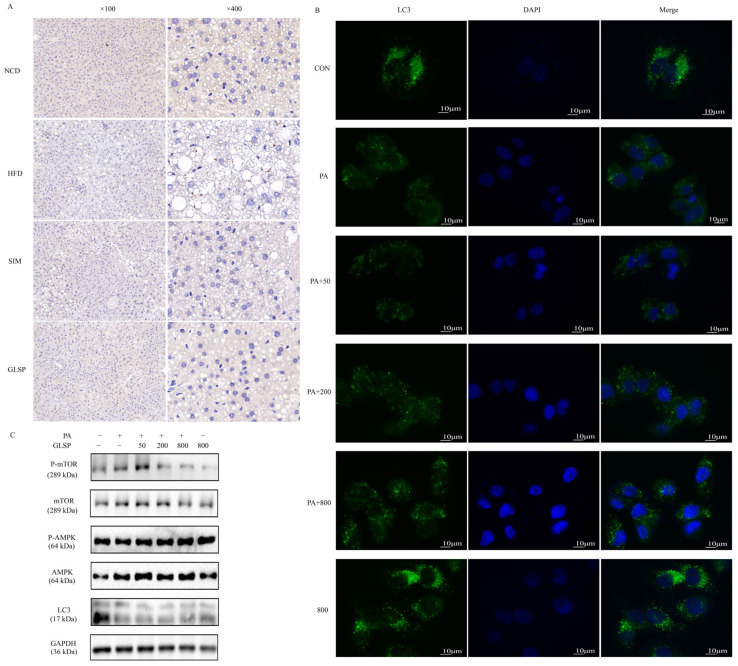
GLSP induced autophagy in vivo and in vitro and activated autophagy via AMPK signal pathway in HepG2 cells. (**A**) Representative images of LC3 immunohistochemistry of liver sections. (**B**) LC3 immunofluorescence staining in HepG2 cells; scale bars, 10 μm. Liver sections or HepG2 cells were stained with anti-LC3 antibody and observed with laser confocal microscope. (**C**) Effects of GLSP on protein expressions of autophagic indicators LC3, P-AMPK/AMPK, and P-mTOR/mTOR in HepG2 cells, with GAPDH as control. HepG2 cells were treated with or without PA (0.6 mM) and with or without GLSP (50, 200, 800 μg/mL).

**Figure 5 antioxidants-13-01501-f005:**
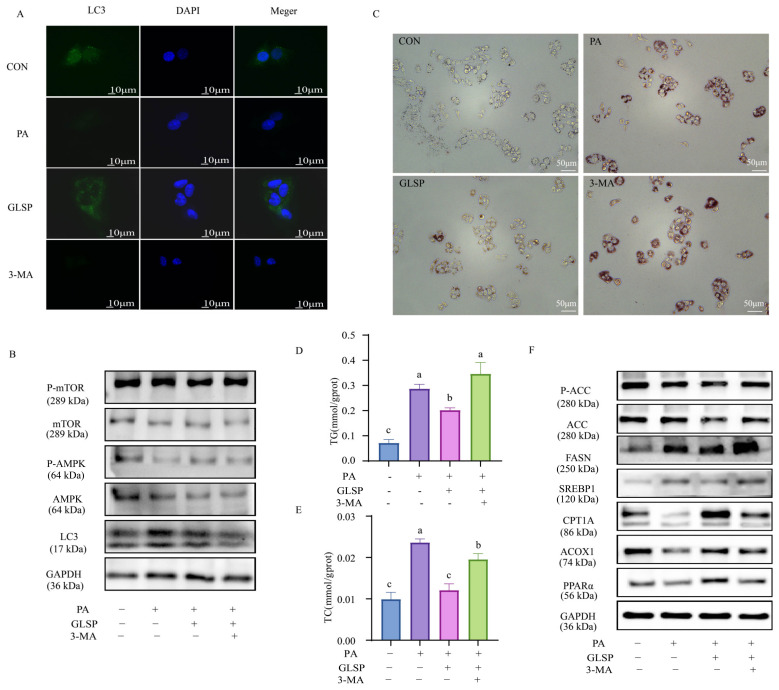
GLSP improved lipid accumulation in HepG2 cells by inducing autophagy. HepG2 cells were treated with 800 µg/mL of GLSP and 3 mM of 3-MA for 12 h and then treated with PA (0.6 mM) for 12 h. (**A**) LC3 immunofluorescence staining showed endogenous LC3 level of HepG2 cells; scale bars, 10 μm. (**B**) Western blot analysis of LC3, P-AMPK/AMPK, and P-mTOR/mTOR protein levels, with GAPDH as control. (**C**) Intracellular lipid accumulation was analyzed by Oil Red O staining; scale bars, 50 μm. (**D**,**E**) Intracellular TG and TC levels in HepG2 cells. (**F**) Western blot analysis of P-ACC/ACC, FASN, SREBP1, ACOX1, CPT1A, and PPARα protein levels, with GAPDH as control. Data are expressed as mean ± SEM, and different letters indicate significant differences (*p* < 0.05). Statistical differences were assessed by Tukey’s test of one-way ANOVA.

**Figure 6 antioxidants-13-01501-f006:**
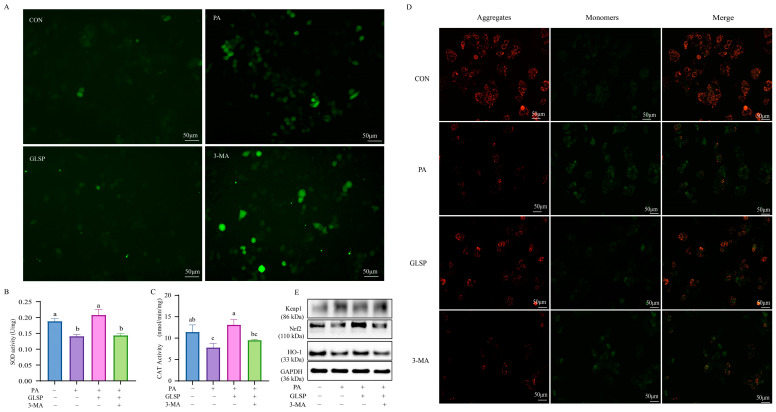
Autophagy contributed to effect of GLSP protecting HepG2 cells from oxidative stress under PA stress. HepG2 cells were treated with 800 µg/mL of GLSP and/or 3 mM of 3-MA for 12 h and then treated with 0.6 mM PA for 12 h. (**A**) Intracellular ROS in HepG2 cells after fluorescence staining is exhibited; scale bars, 50 µm. (**B**,**C**) SOD and CAT were measured with Total Superoxide Dismutase Assay Kit with WST-8 and Micro Catalase (CAT) Assay Kit in HepG2 cells. (**D**) Representative images of JC-1-derived red and green fluorescence of HepG2 cells treated with or without 3-MA; scale bars, 50 µm. (**E**) Effect of GLSP on protein expression of Keap1, Nrf2, and HO-1 protein levels with or without autophagy inhibitor 3-MA. GAPDH was used as control. Data are expressed as mean ± S.E., and different letters indicate significant differences (*p* < 0.05). Statistical differences were assessed by Tukey’s test of one-way ANOVA.

## Data Availability

Data sharing is not applicable to this article as no datasets were analyzed during the current study.

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
