# Peer review of "Ganoderma lucidum Spore Powder Alleviates Metabolic-Associated Fatty Liver Disease by Improving Lipid Accumulation and Oxidative Stress via Autophagy"

_antioxidants, 2024, doi:10.3390/antiox13121501_

Round 1

Reviewer 1 Report

I consider that this is an interesting research that should be accepted for publication in Antioxidants. However the comments previously indicated should be previously dealed with.

Authors report that 12 mice were allocated to each experimental group. Did authors perform replicate or triplicate measurements? 

Did authors identify the polysaccharides and triterpenoids present in GLSP? I consider that this would be and interesting issue.

Authors use a high number of several capitals along the manuscript. In order to facilitate their understanding a table with the explanation of their meanings should facilitate the reading.

In relation to references I consider that more than 60% are from scientists of China or Asian area. In my opinion references from other researchers from other countries should be also considered. 

Reviewer 2 Report

The paper addresses a highly relevant issue, given the rising prevalence of MAFLD and the limited availability of effective treatments.The study explores the potential of GLSP (Ganoderma lucidum extract) in reducing lipid accumulation and oxidative stress, providing a new therapeutic avenue through autophagy activation.

the paper presents promising and well-supported findings on the protective effects of GLSP against MAFLD, backed by a thorough molecular pathway analysis involving AMPK and Keap1/NRF2 in autophagy.

In my opinion LC3 alone is not sufficient to conclusively demonstrate the induction of autophagy. To confirm autophagy induction, it is helpful to include other markers like p62. p62 is degraded during autophagy, so its reduction is often a sign of increased autophagic flux. Conversely, p62 accumulation can indicate blocked autophagy.

The authors use a fluorescent antibody to visualize LC3, but the difference between the control in Figure 4B and that in Figure 5A is unclear to me. In the first, LC3 is localized only in the cytosol, while in the second, it appears in the nucleus. Could the authors clarify this aspect?

I have a few minor queries that I would like to clarify.

  • Figure 3 A and D: Some acquisitions have a very different background color compared to the others (in Fig 3A, samples CTR and 800; in Fig 3D, sample PA monomers). Would it be possible to choose other images?
  • In the caption of Figure 3D, it states, “The nucleus was presented as blue,” but there is no DAPI staining in the image.
  • Figure 5: The Western blot images do not seem to reflect what is described in the text. Specifically, the addition of the inhibitor 3-MA does not appear to have significant effects compared to the condition without it.
  • Line 346: To better interpret LC3 expression, it would be helpful to compare the conditions under examination with each other (or at least the extreme condition with the highest level of GLSP treatment) using an autophagy inhibitor.
  • Line 422: The year of publication is missing in the citation of Ipsen et al.

Round 2

Reviewer 1 Report

Accept in the presente form.

Appropriate to be accepted for publication.